# By Carrot or by Stick: The Influence of Encouraging and Discouraging Facial Feedback on Implicit Rule Learning

**DOI:** 10.3390/bs14010036

**Published:** 2024-01-05

**Authors:** Yiling Liu, Muxin Ouyang, Wenjie Peng, Wenyang Zhang, Keming Lu, Yujun He, Xiangyan Zeng, Jie Yuan

**Affiliations:** 1The School of Psychology, South China Normal University, Guangzhou 510631, China; 2021023704@m.scnu.edu.cn (Y.L.); 2023023873@m.scnu.edu.cn (W.P.); 20182921043@m.scnu.edu.cn (W.Z.); 2021023798@m.scnu.edu.cn (K.L.); 2022023864@m.scnu.edu.cn (Y.H.); zengxy@scnu.edu.cn (X.Z.); 2Psychology Department, Skidmore College, Saratoga Springs, NY 12866, USA; muxinouyang@skidmore.edu

**Keywords:** implicit learning, social feedback, encouraging, discouraging, facial expression

## Abstract

Implicit learning refers to the process of unconsciously learning complex knowledge through feedback. Previous studies investigated the influences of different types of feedback (e.g., social and non-social feedback) on implicit learning. This study focused on the social information presented in the learning situation and tried to explore the effects of different social feedback on implicit rule learning. We assigned participants randomly into an encouraging facial feedback group (happy expression for correct answer, neutral but not negative expression for incorrect answer) and a discouraging facial feedback group (neutral but not happy expression for correct answer, negative expression for incorrect answer). The implicit learning task included four difficulty levels, and social feedback was presented in the learning phase but not the testing phase in two experiments. The only difference between the two experiments was that the sad face used as negative feedback in Experiment 1 was replaced with an angry face in Experiment 2 to enhance the ecological validity of the discouraging facial feedback group. These two experiments yielded consistent results: the performances in the encouraging facial feedback group were more accurate in both the learning and the testing phases at all difficulty levels. These findings indicated that the influence of encouraging social feedback for a better implicit learning achievement was stable and established a new groundwork for future research on incentive-based education, making it critical to investigate the impact of various forms of encouraging-based education on learning.

## 1. Introduction

### 1.1. Implicit Learning

Implicit learning is the incidental, non-episodic acquisition of complex information during which learners are unconscious of what has been learned [1,2]. And it plays a significant role in structuring our perceptions, skills, and behaviors [1,3,4,5]. What makes implicit learning different from explicit learning is that implicit learning is an early evolved cognitive process, which helps humans obtain abstract information and to implicitly acquire knowledge that cannot be expressed verbally [2,6,7,8]. Various paradigms have emerged with the purpose of studying the features and mechanisms of implicit learning, such as artificial grammar learning [1], serial reaction time task (SRTT) [3], visuospatial concept learning [9], statistical learning [4], and frequency and probability learning [7].

Implicit rule learning, as an essential part of implicit learning, has been widely investigated. Previous studies mainly focused on two types of implicit rule learning: sequential learning and non-sequential learning, including the simple recurrent network [10], the competitive chunking model [11], learning non-sequential functions in a video game [12], and learning to recognize if a line drawing conforms to the rules [13].

Most previous researchers focused on exploring the process of implicit learning, paying more attention to learning itself. They also paid less attention to exploring the influence of different types of feedback on the learning process. Recently, exploring the relationship between implicit learning and different types of feedback is becoming a research hotspot in the implicit learning field.

### 1.2. The Social Feedback on Implicit Learning

Feedback refers to the procedures used to inform a learner whether a response is correct or incorrect [14]. Feedback plays a crucial role in improving knowledge and skill acquisition [15]. It has been demonstrated that students could receive feedback to understand their learning progress, weaknesses, and areas for improvement, thus adjusting their learning strategies and methods accordingly [16]. Feedback can be divided into two main types: non-social feedback (feedback is given through symbolic messages, e.g., traffic light icons) and social feedback (feedback is given through social messages, e.g., gestures, facial expressions, body postures). Social feedback provided by a teacher/educator aims to help students transition from a descriptive phase to an interpretative phase, emphasizing analysis, monitoring, and evaluation within the learning process. Particularly, facial feedback was used as social feedback consistently in previous studies comparing the influence of non-social and social feedback on implicit rule learning [17,18,19,20]. Through facial feedback, participants received correct or incorrect messages through different facial expressions (activation of facial muscles).

Previous studies mainly focused on exploring the difference between social feedback and non-social feedback in terms of their impact on implicit learning. Hurlemann et al. and Legaz et al. found that social feedback was more conducive to implicit learning than non-social feedback using happy expression as positive feedback and angry expression as negative feedback [17,20]. Hu et al. replicated the finding of Hurlemann et al. in Caucasian participants but found an inverted effect that non-social feedback is more conducive to implicit learning in Chinese participants [18]. Beston et al. employed an implicit rule learning paradigm in which participants were presented with card triads and were asked to determine whether the card triads complied with a rule. They found that there was no behavioral (accuracy) difference between social feedback and non-social feedback on implicit learning [19]. To sum up, these previous studies obtained inconsistent results about the effect of social and non-social feedback on implicit rule learning.

Recent studies have put more emphasis on the effect of different types of social feedback on implicit learning. Sobczak and Bunzeck investigated the influence of approve (only positive feedback for better performance) and disapprove (only negative feedback for worse performance) social feedback on explicit and implicit evaluative learning. They found that the approve feedback condition was more conducive to implicit evaluative learning than the disapprove feedback condition [21]. Ou et al. explored the in-group and out-group facial feedback in implicit rule learning and indicated that the in-group (East Asian) facial feedback was more conducive to learning than the out-group (Western) facial feedback on low-difficulty tasks [22].

In our view, exploring the influence of different types of social feedback on implicit learning is becoming a new direction.

### 1.3. Carrot and Stick: Encouraging and Discouraging Social Feedback

The proverb “carrot-and-stick” states that one could strike a donkey with a stick or put a carrot in front of it to get it to move forward. Such a motivating strategy involves using both positive and negative reinforcements [23]. The carrot and stick theory has been explored in various fields [23,24,25], such as helping employees maintain a healthy lifestyle and facilitating explicit learning.

Specifically, a study of a company’s incentives for employees to maintain a healthy lifestyle showed that even though stick and carrot policies are formally the same (i.e., coins gained and lost), they did not necessarily deliver the same message to employees. In other words, the stick reflected the company’s negative attitudes [26]. Eskreis-Winkler et al. obtained similar conclusions in educational practices [27]. Researchers gave two groups of participants different types of feedback: one group received feedback only when they answered incorrectly (failure feedback), and the other group received feedback only when they answered correctly (success feedback). Both types of feedback conveyed the correct answer because there were only two-answer options. In other words, both success and failure feedback conveyed information about the correct answer. It turned out that the participants learned more from success feedback than from failure feedback [27]. These findings all seemed to imply that people would be in a better position if they learned from success feedback. However, previous studies have focused on the perspective of explicit learning rather than implicit learning, prompting our investigations to concentrate on implicit learning.

### 1.4. The Current Study: Aims and Hypotheses

This study aims to investigate the effects of success and failure social feedback on implicit learning. On the one hand, we used the paradigm from Beston et al. and Ou et al. to create an implicit rule and explore the influence of different types of social feedback on implicit rule learning at different difficulty levels [19,22]. On the other hand, Eskreis-Winkler et al. found that people had better explicit learning performance under the success feedback condition than under the failure feedback condition [27]. Inspired by these two research lines, our study tried to explore the effects of encouraging versus discouraging facial feedback on implicit rule learning. The participants from the encouraging feedback group would receive a happy face for a correct answer and a neutral but not negative face for an incorrect answer, whereas participants in the discouraging facial feedback group would see neutral but not happy faces for their correct responses and sad or angry faces for incorrect answer. There were two experiments. We took angry expressions as negative feedback in Experiment 2, which was more ecologically valid than taking sad expressions as negative feedback in Experiment 1. The experimental design and its main differences from previous studies are illustrated in Figure 1. The implicit learning task consisted of four levels of difficulty, and we conducted 2 × 4 mixed-design ANOVAs with encouraging and discouraging facial expression feedback conditions as between-group factors and difficulty levels as within-group factors. The main purpose of our study was to investigate whether encouraging and discouraging facial feedback had different effects on implicit rule learning at different difficulty levels with different negative facial expressions. We hypothesized that participants would have better overall performances in implicit rule learning tasks after receiving encouraging facial feedback than those who received discouraging facial feedback regardless of sad (Exp. 1) or angry (Exp. 2) expression. This research aimed to build a deeper understanding of the factors and mechanisms that influence implicit rule learning, thus providing valuable information for teaching and learning areas.

## 2. Methods

### 2.1. Participants

In Experiment 1, 52 students from South China Normal University were randomly assigned to one of two experimental groups. Five participants were excluded due to procedural errors, leaving 47 participants (38 females) for later analysis. During the learning phase, 24 participants (20 females; *M*_age_ = 20.25, *SD* = 1.78) received encouraging facial feedback and 23 participants (18 females; *M*_age_ = 19.70, *SD* = 1.33) received discouraging facial feedback.

In Experiment 2, a total of 56 students from South China Normal University were randomly assigned to one of two experimental groups. As four participants were eliminated due to procedural errors, data from 52 participants (37 females) were used in this study. Among all participants, 26 (18 females; *M*_age_ =19.23, *SD* = 1.31) received encouraging facial feedback, and 26 (19 females; *M*_age_ = 20.19, *SD* = 1.92) received discouraging facial feedback during the learning phase.

The study was approved by the ethical committee of South China Normal University (SCNU-PSY-2021-415).All participants were healthy adults, right-handed, and colorblindness-free, and they volunteered after giving written informed consent. The experiment was single-blind, and participants were unaware of the purpose of feedback conditions. The experiment was completed in the Cognitive and Behavior Laboratory at the School of Psychology, South China Normal University.

### 2.2. Stimuli and Materials

#### 2.2.1. Stimuli

We used materials from Beston et al. [19] and Ou et al. [22] as the stimulus to create an implicit rule and explore the influence of different social feedback types on implicit rule learning at different difficulty levels. All cards had a distinct combination of one to three shapes (S: circle, triangle, square), one of three colors (C: red, green, blue), one of three numbers (N: one, two, three), and one of three fillings (F: empty, hashed, full) that either conformed to or failed to conform to the following rule: a legal combination was a group of cards in which all stimulus dimensions (shape, number, color, and filling) were the same exactly or different totally. For example, if the filling, color, and number of all three cards were completely identical (e.g., filling: full; color: red; number: two), and the shape was different totally (circle, triangle, and square), this combination belonged to the legal condition. Correspondingly, any combination of cards that demonstrated a partial repetition (not completely identical or different, such as the numbers of the three cards being 1, 2, and 2) of any dimension was defined as the illegal condition (see Figure 2). Based on the general difficulty of assessing legality, card triads were further divided into four difficulty levels. Leveling up made figuring out card combinations more challenging. Specifically, in the legal combinations, the number of dimensions of card triads that were not identical at all was incremented sequentially as the difficulty increased, while in the illegal combinations, the number of dimensions that were partially identical was incremented one at a time as the difficulty increased.

The number of potential card combinations varied depending on the level of difficulty. In the learning phase, the weighting of each combination was modified so that each level had an equal probability of being presented throughout the staircase procedure so that participants could learn about combinations from all levels. As part of the testing phase, the ratio of legal to illegal conditions was also set at 1:1, and three blocks of the four demanding stimuli were combined. The presentation of card combinations was accomplished by programming with E-Prime 2.0.

#### 2.2.2. Feedback

The facial feedback materials were chosen from The NimStim Face Stimulus Set (http://www.macbrain.org/resources.htm, accessed on 8 June 2022), which consists of naturally posed photographs (e.g., with hair and make-up) of professional actors (21 years old–30 years old), including East Asians, and a wide variety of emotional faces for each individual [30]. We selected happy, neutral, and sad expressions of actors 15F, 17F, and 18F as facial feedback in Exp. 1 and happy, neutral, and angry expressions of actors 17F, 18F, and 19F as facial feedback Exp. 2. One of the faces (18F in Exp. 1 and 19F in Exp. 2) applied in the practice phase did not appear in the learning phase, so only two female faces appeared in the learning phase.

### 2.3. Procedures

The experiment was designed and run using E-Prime 2.0.

#### 2.3.1. Procedure of Experiment 1

Instruction. Participants were instructed to identify the legality of the card triad as fast as possible in the formal experiments, including the learning and testing phases. Prior to the learning phase, all participants were given a brief presentation of neutral, happy, and sad facial expressions so that they could distinguish them correctly. Eight trials were conducted during the practice phase to familiarize the participants with the experimental procedure.

Learning Phase. Participants engaged in an implicit intentional learning task initially. It began with a “+” fixation point in the center of the monitor for 200 milliseconds. The combination card triads were then randomly selected from a database of card combinations and presented in the middle of a 24-inch CRT monitor (1920 × 1080) with a refresh rate of 100 Hz until a response was received. Participants were instructed to state whether the present combination was “legal” or “illegal” by pressing F or J (sequence balance between participants) on the keyboard. Participants were required to respond as fast as possible.

Legal and illegal combinations had an equal likelihood of presentation in levels of difficulty (1:1). For participants in the encouraging feedback group, once they provided a correct answer, a happy face appeared on the screen; but if they provided an incorrect answer, they saw a neutral facial expression (instead of a sad one). For participants in the “discouraging feedback group”, the feedback was given in a more negative manner, as neutral faces (instead of happy ones) appeared after each correct answer, while sad faces appeared after each incorrect one. The facial feedback image was presented for 750 ms before the next trial began (see Figure 3).

A total of four difficulties (1 to 4) were included, and the complexity was increased one at a time. At each level of difficulty, participants were required to complete five cumulatively correct responses for each legal and illegal condition. In other words, any error would reset the number of correct trials for the current difficulty level and condition to zero.

Testing phase. After completing the learning phase, a mixture of four stimuli of varying degrees of difficulty appeared, with equal probabilities for legal and illegal combinations. In the testing phase, participants were instructed to indicate whether each card triad presented was legal or illegal without receiving any feedback. Participants were required to respond as fast as possible.

Each triad was presented for a maximum of 1500 milliseconds, and a response initiated the next triad presentation of a 500-millisecond fixation cross. There were three blocks, and participants had the opportunity to rest one minute in between each block of 200 trials (see Figure 4).

#### 2.3.2. Aim and Procedure of Experiment 2

This investigation followed the same steps as Experiment 1. The only difference was that the angry face replaced the sad face as the discouraging facial feedback in the discouraging facial feedback condition to enhance the ecological validity in the learning phase.

### 2.4. Data Analysis

Both Experiment 1 and Experiment 2 were analyzed using mixed design ANOVAs in SPSS 26.0 with difficulty (level 1, level 2, level 3, and level 4) as repeated-measures factors and feedback condition (encouraging and discouraging) as a between-group factor. Only correct responses were included in all analyses of response time (RT), and RTs shorter than 200 ms and more than 2.5 standard deviations above the mean of their respective difficulty condition for each participant were excluded. Experiment 1 had a learning phase rejection rate of 1.43% and a testing phase rejection rate of 1.83%. In Experiment 2, the rejection rate for the learning phase was 1.91%, and it was 1.67% for the testing phase. Bonferroni correction was applied to all post hoc comparisons when the main effect was significant.

## 3. Results

### 3.1. Results of Experiment 1

#### 3.1.1. Learning Phase of Exp. 1

Number of trials. At first, an analysis of the total number of trials required for participants to enter the testing phase was conducted. There was a significant main effect of difficulty (*F* (3, 135) = 13.685, *p* < 0.001, *η*^2^*_p_* = 0.233) but no main effect of feedback conditions (*F* (1, 45) = 2.461, *p* = 0.124). The interaction between difficulty level and feedback condition was not significant (*F* (3, 135) = 1.645, *p* = 0.182). We applied the Bonferroni post hoc test for multiple comparisons, and the results showed that the number of trials of level 1 (*M*_level 1_ = 22.596, *SD* = 16.772) was significantly smaller than that of level 2 (*M*_level 2_ = 32.620, *SD* = 18.787, *t* (46) = −3.042, *p* = 0.026, CI 95% = [−19.224, −0.824]). The number of trials of level 2 was also significantly smaller than that of level 3 (*M*_level 3_ = 55.468, *SD* = 36.299, *t* (46) = −3.839, *p* = 0.002, CI 95% = [−39.166, −6.957]). However, there was no significant difference between the number of trials of level 2 and level 4 (*M*_level 4_ = 40.021, *SD* = 27.481, *t* (46) = −1.501, *p* = 0.872, CI 95% = [−21.164, 6.377]) as well as level 3 and level 4 (*t* (46) = 2.257, *p* = 0.148, CI 95% = [−2.944, 34.281]).

Accuracy. The results showed two significant main effects of feedback conditions (*F* (1, 45) = 4.263, *p* = 0.045, *η*^2^*_p_* = 0.087) and levels of difficulty (*F* (3, 135) = 12.828, *p* < 0.001, *η*^2^*_p_* = 0.222). However, no significant interaction between the two variables was found (*F* (3, 135) = 0.618, *p* = 0.604). To be more specific, the accuracy of the encouraging facial feedback was higher than that of the discouraging facial feedback (*M*_encouraging_ = 0.757, *SD* = 0.013; *M*_discouraging_ = 0.718, *SD* = 0.014). Subsequent multiple comparison results (Bonferroni) suggested that the accuracy of level 1 (*M*_level 1_ = 0.822, *SD* = 0.021) was slightly higher than that of level 2 (*M*_level 2_ = 0.756, *SD* = 0.016, *t* (46) = 2.640, *p* = 0.063, CI 95% = [−0.002, 0.134]) and significantly higher than level 3 (*M*_level 3_ = 0.677, *SD* = 0.018, *t* (46) = 5.179, *p* < 0.001, CI 95% = [0.066, 0.223]) and level 4 (*M*_level 4_ = 0.696, *SD* = 0.018, *t* (46) = 4.167, *p* = 0.001, CI 95% = [0.043, 0.207]). The accuracy of level 2 was significantly higher than that of level 3 (*t* (46) = 3.160, *p* = 0.003). No significant difference was found between the accuracy of levels 2 and 4 (*t* (46) = 2.360, *p* = 0.130, CI 95% = [−0.009, 0.128]) and level 3 and level 4 (*t* (46) = −0.952, *p* = 0.363, CI 95% = [−0.078, 0.039]) (see Figure 5).

Reaction time. With regard to the variable of reaction time in Experiment 1, 1.4% of trials were deleted. According to the results from the 2 × 4 mixed design ANOVA, a significant main effect was found for difficulty level (*F* (3, 135) = 6.477, *p* < 0.001, *η*^2^*_p_* = 0.126) but not for feedback condition (*F* (1, 45) = 0.998, *p* = 0.323). No significant interaction between the two was found, either (*F* (3, 135) = 0.156, *p* = 0.926). The results suggested that as difficulty level increased, reaction time increased subsequently. We applied the Bonferroni post hoc test for multiple comparisons, and the results showed that the reaction time of level 1 (*M*_level 1_ = 1061 ms, *SD* = 51 ms) was significantly shorter than that of level 3 (*M*_level 3_ = 1179 ms, *SD* = 49 ms, *t* (46) = −2.817, *p* = 0.043, CI 95% = [−233, −2]) and slightly shorter than that of level 4 (*M*_level 4_ = 1222 ms, *SD* = 68 ms, *t* (46) = −2.538, *p* = 0.088, CI 95% = [−335, 14]). The reaction time of level 2 (*M*_level 2_ = 1071 ms, *SD* = 48 ms) was significantly shorter than that of level 3 (*t* (46) = −3.996, *p* = 0.001, CI 95% = [−182, 33]) and level 4 (*t* (46) = −3.081, *p* = 0.021, CI 95% = [−286, 16]). However, there was no significant difference between the reaction time of level 1 and level 2 (*t* (46) = −0.265, *p* = 1.000, CI 95% = [−111, 92]), and level 3 and level 4 (*t* (46) = −1.110, *p* = 1.000, CI 95% = [−150, 64]).

#### 3.1.2. Testing Phase of Exp. 1

Accuracy. The analysis of accuracy showed a main effect of feedback conditions (*F* (1, 45) = 6.233, *p* = 0.016, *η*^2^*_p_* = 0.122) but no main effect of difficulty (*F* (3, 135) = 0.647, *p* = 0.586). No significant interaction between the two was found, either (*F* (3, 135) = 0.230, *p* = 0.875). The accuracy of the encouraging feedback was higher than that of the discouraging feedback (*M*_encouraging_ = 0.699, *SD* = 0.017; *M*_discouraging_ = 0.637, *SD* = 0.018) (see Figure 5).

Reaction time. The rejection trials accounted for 1.8% of the correct counter-attempts. There was a main effect of difficulty (*F* (3, 135) = 9.785, *p* < 0.001, *η*^2^*_p_* = 0.179) but no main effect of feedback condition (*F* (1, 45) = 0.463, *p* = 0.500). The interaction between the two was not significant (*F* (3, 135) = 1.155, *p* = 0.329). Specifically, in the testing phase of Experiment 1, reaction time decreased as difficulty level increased. According to the result of the Bonferroni post hoc test for multiple comparisons, it was discovered that the reaction times of level 1 (*M*_level 1_ = 780 ms, *SD* = 18 ms) were significantly shorter than those of level 3 (*M*_level 3_ = 807 ms, *SD* = 18 ms, *t* (46) = −4.109, *p* = 0.001, CI 95% = [−46, −9]) and level 4 (*M*_level 4_ = 813 ms, *SD* = 19 ms, *t* (46) = −3.539, *p* = 0.006, CI 95% = [−60, −7]). Additionally, the reaction time of level 2 (*M*_level 2_ = 792 ms, *SD* = 18 ms) was significantly shorter than that of level 3 (*t* (46) = −2.934, *p* = 0.031, CI 95% = [−30, −1]) and level 4 (*t* (46) = −2.986 *p* = 0.027, CI 95% = [−41, −2]). There was no significant difference between the reaction time of level 1 and level 2 (*t* (46) = −2.100, *p* = 0.248, CI 95% = [−28, 4]) and level 3 and level 4 (*t* (46) = −0.977, *p* = 1.000, CI 95% = [−22, 11]).

### 3.2. Results of Experiment 2

#### 3.2.1. Learning Phase of Exp. 2

Number of trials. The data analysis methods used in Experiment 2 were the same as those applied in Experiment 1. With regard to the learning phase in Experiment 2, the analysis of the total number of trials required to succeed showed that there was a significant main effect of difficulty level (*F* (3, 150) = 18.985, *p* < 0.001, *η*^2^*_p_* = 0.275) but no main effect of feedback conditions (*F* (1, 50) = 2.375, *p* = 0.130). No significant interaction between the two variables was found (*F* (3, 150) =0.538, *p* = 0.657). Results from multiple comparisons showed that the number of trials of level 1 (*M*_level 1_ = 19.346, *SD* = 9.758) was significantly smaller than that of level 2 (*M*_level 2_ = 29.731, *SD* = 22.166, *t* (51) = −2.995, *p* = 0.025, CI 95% = [−19.897, −0.872]). The number of trials of level 2 was significantly smaller than that of level 3 (*M*_level 3_ = 53.039, *SD* = 41.160, *t* (51) = −4.097, *p* = 0.001, CI 95% = [−39.089, −7.526]) and level 4 (*M*_level 4_ = 63.308, *SD* = 51.860, *t* (51) = −4.482, *p* < 0.001, CI 95% = [−54.287, −12.867]). However, there was no significant difference between the number of trials of level 3 and level 4 (*t* (51) = −1.212, *p* = 1.000, CI 95% = [−33.678, 13.140]).

Accuracy. The analysis showed that there were two significant main effects of feedback conditions (*F* (1, 50) = 4.301, *p* = 0.043, *η*^2^*_p_* = 0.079) and levels of difficulty (*F* (3, 150) = 24.275, *p* < 0.001, *η*^2^*_p_* = 0.327). However, no significant interaction between the two variables was found (*F* (3, 150) = 0.823, *p* = 0.483). The accuracy of the encouraging facial feedback was higher than that of the discouraging facial feedback (*M*_encouraging_ = 0.762, *SD* = 0.016; *M*_discouraging_ = 0.714, *SD* = 0.016). We applied the Bonferroni post hoc test for multiple comparisons, and the results showed that the accuracy of level 1 (*M*_level 1_ = 0.831, *SD* = 0.021) was significantly higher than that of level 3 (*M*_level 3_ = 0.700, *SD* = 0.019, *t* (51) = 4.679, *p* < 0.001, CI 95% = [−0.054, 0.208]) and level 4 (*M*_level 4_ = 0.630, *SD* = 0.021, *t* (51) = 6.700, *p* < 0.001, CI 95% = [0.119, 0.283]). Moreover, the accuracy of level 2 (*M*_level 2_ = 0.789, *SD* = 0.017) was significantly higher than that of level 3 (*t* (51) = 3.826, *p* = 0.003, CI 95% = [0.024, 0.153]) and level 4 (*t* (51) = 5.889, *p* < 0.001, CI 95% = [0.085, 0.232]). The accuracy of level 3 was significantly higher than level 4 (*t* (51) = 3.333, *p* = 0.009, CI 95% = [0.013, 0.127]). However, there was no significant difference in accuracy between level 1 and level 2 (*t* (51) = 1.680, *p* = 0.574, CI 95% = [−0.026, 0.111]) (see Figure 5).

Reaction time. For Experiment 2, 1.7% of trials were deleted at the reaction time. According to the results from the analyses, there was no significant difference in feedback condition (*F* (1, 50) = 2.544, *p* = 0.117), difficulty level (*F* (3, 150) = 0.959, *p* = 0.414), or interaction (*F* (3, 150) = 0.318, *p* = 0.812).

#### 3.2.2. Testing Phase of Exp. 2

Accuracy. An analysis of accuracy showed both significant main effects of feedback conditions (*F* (1, 50) = 4.902, *p* = 0.031, *η*^2^*_p_* = 0.089) and difficulty levels (*F* (3, 150) = 19.227, *p* < 0.001, *η*^2^*_p_* = 0.278). Yet no significant interaction between the two was found (*F* (3, 150) = 1.157, *p* = 0.328). The accuracy of the encouraging facial feedback was higher than that of the discouraging facial feedback (*M*_encouraging_ = 0.709, *SD* = 0.019; *M*_discouraging_ = 0.650, *SD* = 0.019). More specifically, we applied the Bonferroni post hoc test for multiple comparisons, and the results showed that the accuracy of level 1 (*M*_level 1_ = 0.763, *SD* = 0.021) was significantly higher than that of level 2 (*M*_level 2_ = 0.710, *SD* = 0.015, *t* (51) = 4.000, *p* = 0.001, CI 95% = [0.018, 0.087]), level 3 (*M*_level 3_ = 0.662, *SD* = 0.017, *t* (51) = 4.348, *p* < 0.001, CI 95% = [0.036, 0.164]), and level 4 (*M*_level 4_ = 0.583, *SD* = 0.026, *t* (51) = 4.737, *p* < 0.001, CI 95% = [0.075, 0.285]). Additionally, the accuracy of level 2 was significantly higher than that of level 3 (*t* (51) = 3.429, *p* = 0.009, CI 95% = [−0.009, 0.087]) and level 4 (*t* (51) = 4.233, *p* < 0.001, CI 95% = [0.046, 0.209]). The accuracy of level 3 was significantly higher than level 4 (*t* (51) = 3.950, *p* = 0.001, CI 95% = [0.025, 0.134]). In summary, the accuracy gradually decreased as the difficulty level of learning increased (see Figure 5).

Reaction time. For the testing phase in Experiment 2, the rejection trials accounted for 1.7% of the correct counter-attempts. There was a main effect of difficulties (*F* (3, 150) = 22.440, *p* < 0.001, *η*^2^*_p_* = 0.310) but no main effect of feedback conditions (*F* (1, 50) = 0.013, *p* = 0.909). The interaction between the two variables was insignificant (*F* (3, 150) = 0.920, *p* = 0.433). Still, as the difficulty increased, the reaction time gradually increased. We applied the Bonferroni post hoc test for multiple comparisons, and the results showed that the reaction time of level 1 (*M*_level 1_ = 776 ms, *SD* = 20 ms) was significantly shorter than that of level 2 (*M*_level 2_ = 801 ms, *SD* = 20 ms, *t* (51) = −5.750, *p* < 0.001, CI 95% = [−37, 13]), level 3 (*M*_level 3_ = 811 ms, *SD* = 21 ms, *t* (51) = −5.447, *p* < 0.001, CI 95% = [−53, 18]), and level 4 (*M*_level 4_ = 830 ms, *SD* = 22 ms, *t* (51) = −5.680, *p* < 0.001, CI 95% = [−81, 28]). The reaction time of level 2 was significantly shorter than that of level 4 (*t* (51) = −3.830, *p* = 0.002, CI 95% = [−51, −8]). The reaction time of level 3 was significantly shorter than that of level 4 (*t* (51) = −2.861, *p* = 0.037, CI 95% = [−38, −1]). However, there was no significant difference in reaction time between level 2 and level 3 (*t* (51) = −2.389, *p* = 0.124, CI 95% = [−22, 2]).

## 4. Discussion

Our study aimed to examine the impact of encouraging and discouraging facial feedback on implicit rule learning using the card triad rule learning paradigm. The most notable finding was that encouraging facial feedback had a significantly more positive effect on implicit rule learning compared to discouraging facial feedback. We assigned participants randomly into the encouraging facial feedback group (i.e., happy expression for correct answer, neutral but not negative expression for incorrect answer) and discouraging facial feedback group (i.e., neutral but not happy expression for correct answer, negative expression for incorrect answer). Social feedback was presented in the learning phase but not the testing phase in two experiments. The implicit learning task consisted of four levels of difficulty, which were presented sequentially and in ascending order during the learning phase. During the testing phase, a mixture of all four difficulty levels was presented in three blocks. In Experiment 1, sad faces were used as negative facial feedback. The results showed a higher accuracy in both the learning and testing phases for the encouraging facial feedback group. In Experiment 2, the only difference from Experiment 1 was the replacement of sad faces with angry faces to enhance the ecological validity in the discouraging facial feedback group. The results reinforced our expectations, with the encouraging facial feedback group achieving a higher accuracy than the discouraging facial feedback group in both the learning and testing phases. Overall, the results from both experiments consistently indicated that encouraging facial feedback led to better outcomes in implicit rule learning.

We attempted to discuss the possible underlying mechanisms from the perspectives of attentional bias of negative facial feedback and goal-setting theory. From the perspective of attentional bias, one of the latest studies found that attention could influence implicit learning [31]. In the area of implicit rule learning, Ou et al. employed the relationship between attention and implicit learning to explain their finding that the accuracy of the implicit learning task of an out-group feedback condition was less than that of the in-group feedback condition at the least difficulty level. In their view, individuals in the out-group facial feedback condition paid more attention to out-group feedback and then paid less attention to the implicit learning task because out-group faces are more threatening than in-group ones. Consequently, the performance of implicit learning in the out-group feedback condition was worse than that of the in-group feedback condition [22]. Similarly, since previous studies found that negative stimulus attracts more attention [32], individuals in the discouraging facial feedback condition (negative feedback for incorrect answers and neural feedback for correct answer) may pay more attention to the feedback and then pay less attention to the implicit learning task. Consequently, the current study found that the performance of the implicit learning of the discouraging facial feedback condition was worse than that of the encouraging facial feedback condition (positive feedback for correct answer and neural feedback for incorrect answer). Conversely, individuals in the encouraging feedback condition may pay less attention to the encouraging facial feedback and then pay more attention to the task itself. According to goal-setting theory [33], dissatisfaction with a past task performance could lead individuals to increase goal-setting and, subsequently, to improve task performance in the encouraging feedback condition. Thus, the attention bias of a negative feedback stimulus in the discouraging feedback condition and increased goal-setting in the encouraging feedback condition may be the possible underlying mechanisms of the encouraging feedback advantage in implicit rule learning.

### 4.1. The Relationship with Previous Related Studies

Our study found that encouraging facial feedback was more conducive to learning than discouraging facial feedback, and previous studies had similar results on the effects of performance from positive and negative perspectives. For instance, Eskreis-Winkler et al. found that participants learned more from success feedback than from failure feedback; in other words, failure undermined learning [27]. A study with nine-month-old infants came to a similar conclusion: boosting positive emotions and possibly down-regulating negative emotional responses might be crucial for improving performance and learning complicated manipulation skills in infancy [34]. Additionally, Frank et al. discovered that Parkinson’s patients treated with dopamine medication were more sensitive to positive results than negative ones [23]. Similar to a previous study that success feedback only was more conducive to explicit learning than failure feedback only, our experiment found a positive effect of encouraging feedback from the perspective of implicit learning.

### 4.2. Ecological Validity: The Sad-to-Angry Shift in Discouraging Facial Feedback from Exp. 1 to Exp. 2

Sad and angry expressions were two main types of negative facial feedback used by previous studies that investigated the relationship between implicit learning and social feedback. For example, Beston et al. and Ou et al. employed a sad expression as negative feedback, and Hurlemann et al., Hu et al., and Legaz et al. employed an angry expression as negative feedback [17,18,19,20,22]. Since angry expressions were more aggressive than sad expressions, sad expressions were provided for incorrect answers in several situations such as the Internet environment. In other words, sad expressions as negative feedback did not completely reflect the nature of educational reality. For example, when students perform worse in some learning tasks, their teachers or parents tend to express an angry emotion more frequently than a sad emotion to them in daily life. Thus, based on the above deduction, we took angry expressions as negative feedback in Exp. 2, which is more ecologically valid than sad expressions as negative feedback in Exp. 1.

### 4.3. Encouraging and Discouraging Feedback in Relation to Positive and Negative Reinforcement on Neo-Behaviorism

Neo-behaviorist B.F. Skinner proposed the concept of “reinforcements”, referring to the phenomenon that either providing a reward or removing a punishment would increase the frequency of certain behaviors. The former is called “positive reinforcement”, while the latter is named “negative reinforcement”.

As a possible analogy of our findings with Skinner’s positive and negative reinforcements, we inferred that in the encouraging facial feedback group, positive facial feedback reinforced an implicit learning behavior by creating anticipation for more happy faces. On the other hand, in the discouraging facial feedback group, negative facial feedback reinforced an implicit learning behavior by creating tendencies of avoiding sad or angry faces.

Put it simply, in our experiment, the encouraging facial feedback could correspond to the positive reinforcement, whereas the discouraging facial feedback group could correspond to the negative reinforcement.

A previous study found that positive reinforcements increased task accuracy compared to negative reinforcements [35]. The present study yielded similar findings, i.e., the accuracy of implicit learning of the encouraging facial feedback group was higher than that of the discouraging facial feedback group. Thus, this possible relationship between our findings and Skinner’s reinforcements is explicable.

### 4.4. Limitations and Future Directions

Our research had several limitations which should be discussed. Firstly, for each experiment, we used only two East Asian female faces with different emotions as feedback material. This was similar to the study by Hu et al., which also used only female faces for social feedback [18]. Although the educators (e.g., secondary school teachers) were mostly females, East Asian male faces could also be added as one of the feedback messages in subsequent studies. Using a wider variety of face pictures (e.g., gender) would make the results more representative. Secondly, most of the participants in our study were female college students. Therefore, it was unknown whether the unbalanced gender ratio affected the results of implicit learning based on social feedback, and future research could balance this quantitative difference. Thirdly, the sample size was not representative enough to generalize the conclusions to a country with a vast population and cultural diversity. Expanding the number and diversity of participants in terms of geographical location, age, and educational level would be worthwhile for future research. Fourthly, our study was a single-blind experiment in which the experimenter was the experimental designer, which, to some extent, affected the experimental manipulation. Therefore, a double-blind experiment could be used for future research. Fifthly, we did not explore participants’ learning curves. Future studies can analyze participants’ learning curves during the learning period and compare the learning curves of different types of social feedback. Sixthly, we did not evaluate the individuals’ explicit knowledge of this implicit rule learning task. Future studies can add an evaluation phase to check the implicit learning state.

Although previous research provided strong evidence of the positive effects of encouraging facial feedback on learning outcomes, the underlying behavioral and neural mechanisms still require further exploration. Beston et al. used EEG technology, such as event-related potentials (ERPs), to obtain unbiased evidence of implicit learning [19]. Future studies could also explore the mechanisms of different social feedback on implicit learning using EEG technology. From the aspect of neo-behaviorism, the positive and negative reinforcements corresponding to encouraging and discouraging feedback, respectively, could be presented through the Skinner box. Future research can explore the effect of neo-behaviorism on different types of social feedback by allowing the two groups of participants to make their own choices after understanding how the Skinner box works. Another notable point was that a previous study showed that due to the nature of adolescents’ brain development, their cognitive functions not being fully developed, their recognition of emotions was different from that of adults [36]. Thus, future research could expand the group of participants to longitudinally compare the differences between encouraging and discouraging facial feedback in different age groups.

### 4.5. Practical Implications

Our findings have important practical implications. Our study investigated the effectiveness of encouraging and discouraging facial feedback on implicit rule learning and discovered that encouraging feedback was more conducive to implicit rule learning than discouraging feedback. In the comparison between the sayings “success is the mother of success” and “failure is the mother of success,” the current study provided evidence to support the former saying, which is also consistent with the findings of Eskreis-Winkler and Fishbach [27]. Put it simply, encouraging facial feedback (positive reinforcement or the so-called “carrots”) was more effective in implicit rule learning compared to discouraging facial feedback (negative reinforcement or the so-called “sticks”). The findings help us to extend the effect of encouraging and discouraging feedback on learning from explicit learning tasks to the area of implicit learning [27], indicating a robust effect of encouraging feedback during the process of learning. Taken together, the combination of previous studies’ and the current study’s findings suggests that providing encouraging feedback to young adult learners is more conducive to their learning than providing discouraging feedback. Additionally, the result of this study might support the reality of modeling effect, i.e., learners’ perceived expectation from social feedback might influence their performance itself. When receiving encouraging facial feedback, learners tend to perceive the expectation as lower, and tolerance as higher. Therefore, they might feel less stressed and then tend to perform better. When learners receive discouraging facial feedback, they perceive the expectation as higher, and tolerance as lower. Therefore, they might feel more stress and then tend to perform worse. Thus, for educators, lowering excessive expectations and increasing tolerance and acceptance towards learners is more conducive to their learning. Furthermore, our study lays a foundation for future research on incentive-based education, emphasizing the need to explore the impact of different forms of encouraging-based education on learning outcomes.

## 5. Conclusions

This study aimed to examine the differential effects of encouraging facial feedback versus discouraging facial feedback on implicit rule learning. Two experiments that employed sad and angry expressions as negative feedback yielded consistent results: the performances from the encouraging facial feedback group showed a higher accuracy in both the learning and the testing phases at all difficulty levels. These findings indicated that encouraging social feedback displayed a more stable positive effect on implicit rule learning than discouraging social feedback.

## Figures and Tables

**Figure 1 behavsci-14-00036-f001:**
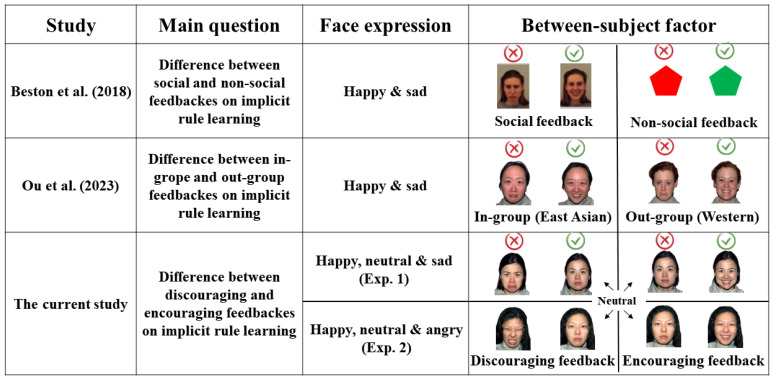
The main differences between the current study and two previous relevant studies. The previous studies focused on exploring social and non-social feedback or in-group and out-group feedback and their effects on implicit rule learning with happy and sad face expressions for correct and incorrect answers. The current study focused on the effect of encouraging (neutral expression for incorrect answer and happy expression for correct answer) and discouraging (neutral expression for correct answer, sad expression for incorrect answer in Exp. 1, and angry expression for incorrect answers in Exp. 2) feedback on implicit rule learning. Beston et al.’s [19] social feedback materials were collated from The Karolinska Directed Emotional Faces database [28,29]. The current study and Ou et al. [22] both used The NimStim Face Stimulus Set to compose facial feedback materials [30].

**Figure 2 behavsci-14-00036-f002:**
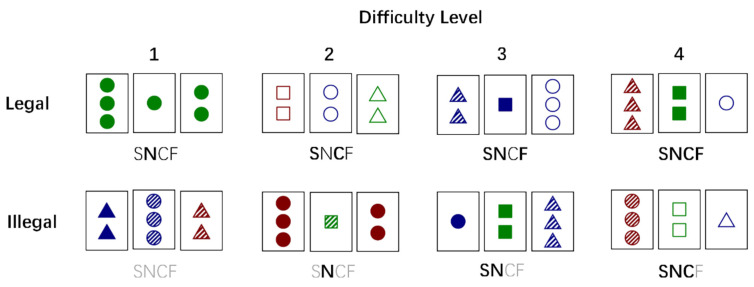
Examples of legal and illegal card triads at each level of difficulty. In the legal combined cards, completely different latitudes are added with increasing difficulty. In the illegal ones, as the difficulty increases, parts of the same latitude numbered one to four match the difficulty level from four to one. The code under each triad represents the composition type of the four dimensions: thin black letters indicate that the dimension is identical in all three card combinations, black bold letters indicate that the dimension is entirely different, and grey letters indicate that the dimension is partially different or the same.

**Figure 3 behavsci-14-00036-f003:**
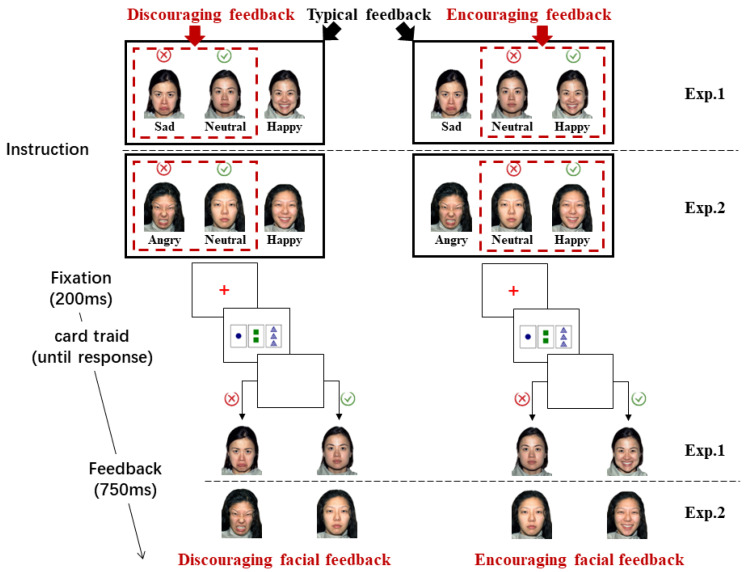
Structure of the learning phase trials. For participants in discouraging feedback group (see the red square on the **left**), once they provided an incorrect answer, a sad/angry face was displayed on the screen; but if they provided a correct answer, they saw a neutral facial expression (instead of a happy one). For participants in encouraging feedback group, on the other hand, the feedback was given in a more positive manner (see the red square on the **right**) as neutral faces (instead of sad ones) displayed after each incorrect answer, while happy faces were displayed after each correct one. Both groups of participants received either discouraging (**left**) or encouraging (**right**) feedback in the learning phase. In Experiment 1, the expression “sad” was used as negative feedback, while in Experiment 2, it was replaced by the expression “angry”.

**Figure 4 behavsci-14-00036-f004:**
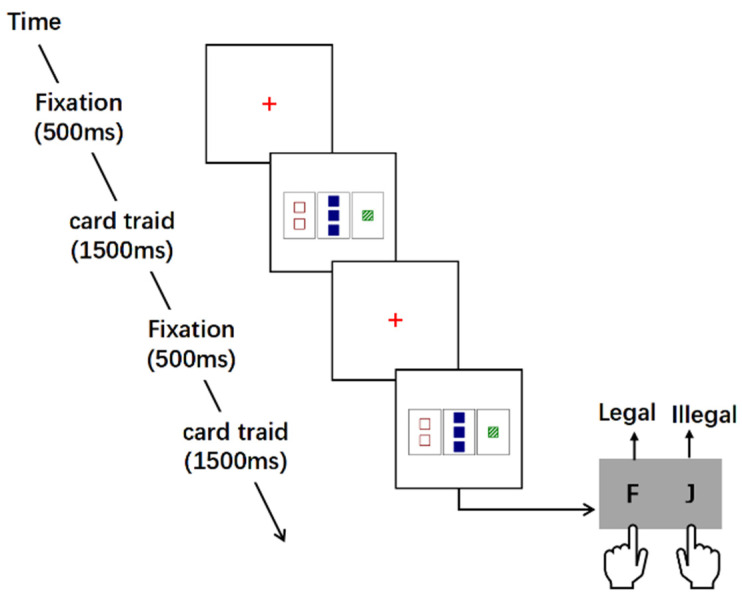
Trial structure of the testing phase. Participants were required to respond at every trial. Response side counterbalanced across participants.

**Figure 5 behavsci-14-00036-f005:**
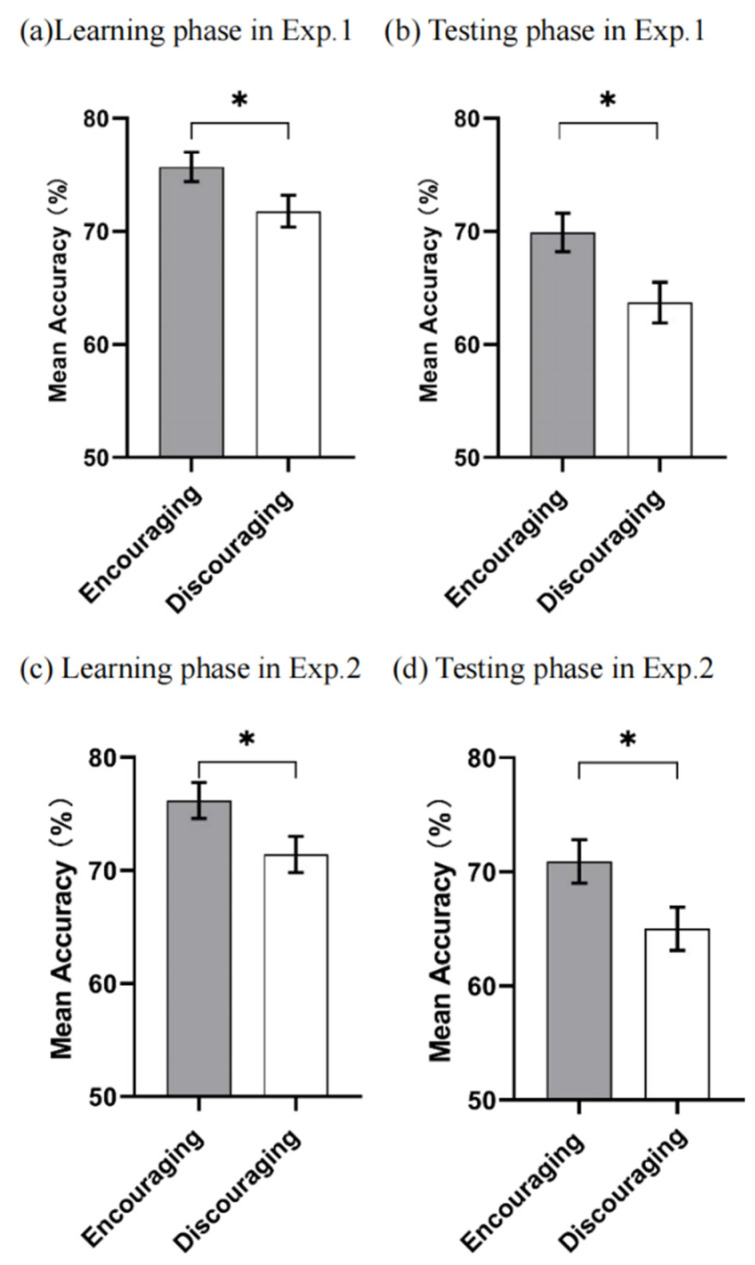
The results of Experiments 1 and 2, featuring the effect of feedback conditions on mean accuracy. Error bars represent SEM. * *p* < 0.05.

## Data Availability

The data presented in this study are available on request from the corresponding author. The data are not publicly available due to privacy concerns mentioned in the IRB protocol.

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
