# Peer review of "By Carrot or by Stick: The Influence of Encouraging and Discouraging Facial Feedback on Implicit Rule Learning"

_behavsci, 2024, doi:10.3390/bs14010036_

Round 1

Reviewer 1 Report

Comments and Suggestions for Authors

Thank you for giving me an opportunity of reviewing the paper. The paper is well written and address an important challenge in student learning. I still feel you need to mention that in what ways your work is different from Beston et al (2018). I would suggest to incorporate a figure showing the difference in your work and Beston et al (2018).

Also, the concept of Legal and illegal cards is not very clear. Please explain the reason of using the stimulus. 

Lastly, please check the citation style, in few places you have used APA and others are IEEE. Please be consistent

Comments on the Quality of English Language

It needs some minor editing.

Reviewer 2 Report

Comments and Suggestions for Authors

Thank you very much for allowing me to review this manuscript. The study is presented in an excellent manner. Nevertheless, I have a few comments that might enhance the quality of the paper:

In the theoretical part of the article, the term feedback should be precisely defined. In addition, there is no overview of studies that prove the correlation between feedback and performance or the influence of feedback on the learning of individuals.

At the end of the theoretical section, it is not clear which research gap the authors wish to address. How can the relevance of the study be derived from the work carried out to date? It is also necessary to formulate clear research questions and/or research hypotheses, which the authors address in the results and discussion sections. 

The results of the study should be presented in tabular form. Firstly, this makes the text easier to read, and secondly, it makes it easier to view the findings.

The practical implications of the study could be worked out more clearly: What do the findings mean for young adult learning?

Comments on the Quality of English Language

/

Reviewer 3 Report

Comments and Suggestions for Authors

The Authors investigate the differences between positive and negative feedback for implicit rule learning, and found better performances for positive feedback.

1) First of all, the language is sloppy and sometimes difficult to understand, because of several grammar/syntax mistakes, incorrectness and naivety (subjects missing or switched in the middle of a sentence, inappropriate use of -ing forms, etc). I mention the language as the first problem, as at times it makes the meaning of the study almost incomprehensible.

Just a few examples (too many problems to mention them all):

- the first sentence of Introduction (lines 31-34) should be entirely changed into something similar to, e.g.: “Implicit learning is the incidental, non-episodic acquisition of complex information by subjects who are not conscious of what they have learned [1,2]. Compared to explicit learning, implicit learning is a process which started early during evolution, it produces abstract information, and involves implicitly acquiring knowledge that cannot be expressed verbally.”

- Also, line 35: “has gained a great deal of exploration” does not make any sense, it should probably be, e.g.: “has been widely investigated”, or similar.

- The following sentence (lines 36-39) is extremely obscure, e.g., what is the meaning of “from sequential and non-sequential structures”?

- Lines 64 and 66: “false” should be “wrong” or “incorrect”

- Repeatedly in the text: “responses were provided”, “responses were given with”, and “responses were shown” are incorrect sentences

- Lines 392-395 are unintelligible. Also, level is lever?  skinner is Skinner?

…and so on.

2) Lines 101 and 103: the reasons for choosing sad or angry expressions as negative feedback should be better explained; especially puzzling is the fact that frowning is mentioned as negative feedback in several studies. Namely, why is anger more ecological? why is sadness more corresponding to symbolic feedback? the Authors should provide appropriate references. The same insufficient explanation with almost exactly the same words is provided on Lines 221-223.

3) The Figures are not cited in the text of the paper, apart from Figure 1; they should be cited in the appropriate section of Methods or Results

4) Figure 2 is not very effective: the red squares in the Instruction rows are not well explained and are confusing, on Lines 178 and 181 the two similar sentences “(show as a left/right arrow)” are unclear, and the arrows are very small and difficult to see.

5) Figure 4 legend should be simplified: the sentence “The effect of feedback conditions on the mean accuracy in” should be written only once, and only the different conditions should be specified as A), B), etc.

6) The section on Reaction times (Lines 262 and following) is especially puzzling. “As the difficulty increased, the reaction time gradually increased.”: however, in the following lines, RTs for level 1 are said to be slower than for more difficult level! in turn, this seems at odds with the numbers (what’s the unit for these numbers?) given in parentheses. It would help to have a figure for these results- Same problems for lines 280 and following, and for lines 323 and following.

7) throughout Results, the Authors should not write about main effect of interaction: it is either a main effect of a variable, or the effect of the interaction of the two variables

8) Section 4.3 contains several obscure sentences, and other sentences are a repetition of the results: it should be totally rephrased

9) Line 444: it is not clear why the present results are “Contrasting” with those about explicit learning, rather, they appear to extend previous results: the Authors should either clarify or rephrase

Comments on the Quality of English Language

The language is sloppy and sometimes difficult to understand, because of several grammar/syntax mistakes, incorrectness and naivety (subjects missing or switched in the middle of a sentence, inappropriate use of -ing forms, etc). I mention the language as the first problem, as at times it makes the meaning of the study almost incomprehensible.

Just a few examples (too many problems to mention them all):

- the first sentence of Introduction (lines 31-34) should be entirely changed into something similar to, e.g.: “Implicit learning is the incidental, non-episodic acquisition of complex information by subjects who are not conscious of what they have learned [1,2]. Compared to explicit learning, implicit learning is a process which started early during evolution, it produces abstract information, and involves implicitly acquiring knowledge that cannot be expressed verbally.”

- Also, line 35: “has gained a great deal of exploration” does not make any sense, it should probably be, e.g.: “has been widely investigated”, or similar.

- The following sentence (lines 36-39) is extremely obscure, e.g., what is the meaning of “from sequential and non-sequential structures”?

- Lines 64 and 66: “false” should be “wrong” or “incorrect”

- Repeatedly in the text: “responses were provided”, “responses were given with”, and “responses were shown” are incorrect sentences

- Lines 392-395 are unintelligible. Also, level is lever?  skinner is Skinner?

…and so on.

Reviewer 4 Report

Comments and Suggestions for Authors

In the current study, the influence of social feedback on implicit learning was examined. In two experiments, the authors demonstrate that encouraging facial feedback yielded better performance than discouraging facial feedback in an implicit rule-learning task. The manuscript is well-written, yet I have several comments:

1) It is not clear from the text if participants were instructed to respond as fast as possible. If not, it is unclear if the RT measures are valid measures of performance.

2) Was there a difference in the number of trials required to succeed at each level as a function of difficulty or feedback type? It is not clear if learning itself was affected by the feedback manipulation or if there is a general effect of feedback type on performance. Is there a way to examine participants’ learning curve during the learning period and compare their learning curve?

3) Could it be that changes in attention between the two conditions are responsible for this effect? Negative stimuli have been shown to delay responses compared to neutral ones.

4) Was explicit knowledge evaluated? Is it clearly an implicit learning task, or could participants learn the rule explicitly? I wonder if the specific rule used is really difficult to verbalize.

5) Performance was not compared to non-social feedback cues, so it is difficult to conclude something clear regarding the beneficial role of positive social feedback compared to non-social feedback.  

Reviewer 5 Report

Comments and Suggestions for Authors

Previous note

This article holds the potential to be published after some alterations to the original text. The addressed topic sparks a profound interest in research in general and, more specifically, in the field of Educational Psychology, particularly for those devoted to the study of the educational potentials inherent in investigating the differential effects of encouraging facial feedback as opposed to discouraging facial feedback in the implicit learning of rules. Facial expression, as an involuntary behavior, proves capable of facilitating the transmission of relevant non-verbal information in different social contexts. The process of recognizing emotional facial expressions has piqued the curiosity and enthusiasm of various researchers. The validity and application of such studies as tools in various modalities, such as artificial intelligence, training, and individual treatment, prove valuable. Observing this facial feedback in a university context constitutes a motivational factor in content learning, enhancing peer interaction and class dynamics, besides aiding in the development of reasoning and strategies. As an educational resource, this feedback may contribute to the affective, motor, and cognitive dimensions of the student's personality. In practice, the recognition of emotions through facial expressions can be beneficial for improving interpersonal communication, believing that the visualization and reproduction of specific facial expressions can elicit emotional reactions.

In this sequence, the Behavioral Sciences (BS), through the publication of this article, will have the opportunity to contribute to enriching the theoretical foundation of the studied topic and promote the development of similar intervention programs in higher education institutions. Furthermore, it is crucial that the results obtained in this research are adequately clarified and substantiated, with the aim of adding value to the analyzed text. It should be emphasized that the content of this article is intended for a diverse academic audience with varying knowledge levels on the studied topic and should be structured to enhance readers' understanding.

Therefore, it becomes essential for the production of BS, as a scientific journal, to maintain its quality standards in the studied area. In this regard, some suggestions for changes are presented, namely:Parte superior do formulárioParte superior do formulário

 Abstract

 1. Introduction

1.1. Implicit Learning

- The theoretical model serving as the basis for the study should be explicitly stated.

-  Elaborate further on the concept of implicit learning and provide references to additional studies.

1.2. The Social Feedback on Implicit Learning

- Expand further on the concept of social feedback, allowing for a progressive observation of development within the formative context. In other words, through the feedback provided by the teacher/educator, the aim is for students to transition from a descriptive phase to an interpretative phase, emphasizing analysis and involving increased monitoring and evaluation of specific parameters within the learning process.

- Provide a clearer explanation of what facial feedback entails. Our emotions manifest in various ways through our physicality, particularly in facial expressions. The facial feedback theory asserts that the brain receives sensory information through the activation of facial muscles, leading to emotional experiences in the individual.

1.3. Carrot and Stick: Encouraging and Discouraging Social Feedback

- Provide studies that exemplify the applicability of "Carrot and Stick" strategies. The findings from these studies can aid in comprehending the subsequent results and are valuable in identifying how these incentive and punishment strategies manifest in educational practices.

1.4. The Current Study: Aims and Hypotheses 

- The validity of replacing the methodology used by Boston et al. (2028) with the one employed in the study is not very clear. A more robust clarification is needed to validate the results effectively. There is a feeling that the methodology used has a highly experimental nature.

2. Methods  

2.1. Participants  

- Justify the reason for the sample consisting only of individuals of the female gender. What is the importance of indicating that participants are right-handed?

- Additionally, there is information in this subsection that belongs to the procedures. Also, specify that individuals participated voluntarily in the study. Mention the location of the study and that authorization was obtained from the institution's management.

2.2. Stimuli and Materials

2.2.1. Stimuli

- Provide a clearer operationalization of the game and outline its objectives more explicitly.

2.2.2. Feedback 

- For practical reasons, instead of presenting the link, provide a more detailed explanation of the operationalization of the NimStim program.

Parte superior do formulário

2.3. Procedures  

2.3.1. Procedure of Experiment 1 

2.3.2. Aim and Procedure of Experiment 2

2.4. Data analysis
- The process of tallying various response types and their integration into the program is not elucidated.

- To enhance the readability and interpretation of the obtained results, it is advisable to mention that, in statistical analysis, the Bonferroni correction is one of several methods used to address the issue of multiple comparisons. Additionally, it is helpful to clarify that the confidence interval (CI) is an observed range, calculated from observations, and may vary depending on the samples.

3. Results

3.1. Results in Experiment 1

3.1.1. Learning Phase 

3.1.2. Testing Phase 

3.2. Results in Experiment 2

3.2.1. Learning Phase

3.2.2. Testing Phase

4. Discussion  

- It would be advisable to discuss the results by comparing and justifying them with studies from other authors.

4.1. The Relationship to Previous Related Studies

4.2. Ecological Validity: the Sad-to-Angry Shift in Discouraging Facial Feedback from Exp. 1 to 367 Exp. 2  

4.3. Encouraging and Discouraging Feedback in Relation to Positive and Negative Reinforcement on Neo-behaviorism  

Parte superior do formulário

 4.4. Limitations and Future Directions 

- Another limitation is the inability to generalize the conclusions due to the sample size not being representative of a country with a vast population and cultural diversity.

4.5. Practical Implications

- It could be mentioned whether the results of the present study allowed us to verify the reality of the modeling effect of feedback on the perception of third-party expectations about performance.

- The study results suggest that individual reactions to feedback are mediated by self-assessment mechanisms. According to Edwin Locke's Goal-setting Theory, dissatisfaction with past performance should lead to an increase in personal goals set by individuals and subsequently to improved performance. Consequently, it was expected that the feedback signal would not only have a direct effect on performance but also an indirect effect through its influence on the level of satisfaction and the personal goals of the individual. Given the results obtained, an association around this issue could be explored.

5. Conclusion  

Reference

-  It is suggested that, upon reviewing the article, references to more recent studies be included.
